# Resistance towards Critically Important Antimicrobials among *Enterococcus faecalis* and *E. faecium* in Poultry Farm Environments in Selangor, Malaysia

**DOI:** 10.3390/antibiotics11081118

**Published:** 2022-08-18

**Authors:** Sakshaleni Rajendiran, Yuvaneswary Veloo, Syahidiah Syed Abu Thahir, Rafiza Shaharudin

**Affiliations:** Environmental Health Research Centre, Institute for Medical Research (IMR), Ministry of Health, 40170 Shah Alam, Malaysia

**Keywords:** Enterococci, environment, antibiotic resistance, multidrug resistance, poultry

## Abstract

Multidrug resistant (MDR) enterococci pose significant public health challenges. However, the extent of resistance in the environment is less explored. This study aimed to determine the antibiotic resistance in a poultry farm environment. Eighty enterococcal isolates recovered from the soil and effluent water of 28 poultry farms in Selangor state were included in the study for further bacterial identification and antibiotic susceptibility testing using a VITEK 2 system. Data were analyzed using Statistical Package for Social Science (SPSS) version 27. The resistance rate and MDR of enterococcal isolates were reported. Out of 80 isolates recovered, 72 (90%) exhibited resistance to at least one antibiotic, with 50 isolates (62.5%) being found to be MDR. All linezolid-resistant enterococci (LRE) exhibit MDR, which constituted 40% of resistance among all the isolates recovered from poultry environment. Since linezolid is listed as critically important antibiotics for clinical use by the World Health Organization (WHO), the higher resistance towards it and other critically important antibiotic for human use is a serious concern. Hence, relevant agencies need to investigate the use of clinically important antimicrobials in poultry farms paying special attention towards linezolid or any other antibiotics that can facilitate the development of LRE.

## 1. Introduction

On account of the ability to develop resistance towards multiple antibiotics, thus posing a therapeutic challenge and worsening health outcomes among infected individuals, enterococci attract a great deal of attention among clinicians and researchers [1,2]. Hence, it has been enumerated on the global priority pathogen list known as ESKAPE (*Enterococcus faecium, Staphylococcus aureus, Klebsiella pneumoniae, Acinetobacter baumannii, Pseudomonas aeruginosa, Enterobacter* spp.) [3]. Its soaring resistance is of global concern, and mitigation efforts will require collaboration among mulptile agencies. Human, animal, and environmental health are interconnected with respect to the dissemination of antibiotic resistance. The environment plays a crucial role in the indirect dissemination of antibiotic resistance genes, as soil and water can act as reservoirs for pools of diverse bacteria [4].

Since antibiotics can be excreted through urine or feces in an unmetabolized manner, depending on the pharmacokinetics of the drug, antibiotic pollution of the environment may lead to the development of resistance among the microbial population [4,5]. The genetic determinants in resistant bacteria can be transferred at the intra- and interspecies level and may further exacerbate the threat to human and animal health [6,7]. The dilemma is whether resistance arising in an environment of animal production could give rise to further challenges in the management of infectious disease in clinical settings, as there is the risk that transmission of these resistant strains to humans may occur either through the food chain or through occupational exposure. Other modes of spread to the community or environmental exposure can also occur through run-off from farms to the surrounding vicinity [1,2,8,9]. 

Vancomycin-resistant enterococci (VRE) are considered a major concern from a clinical perspective, and have led to the development of newer drugs that have become available, such as linezolid, tigecycline, daptomycin, and the fifth generation of cephalosporin, for treatment of VRE-related infection. However, the emergence of linezolid-resistant enterococci (LRE) is a serious public health concern, as clinicians will be left without treatment options [10,11].

Antibiotic resistance among enterococci is a touchstone representing resistance among Gram-positive bacteria, and the high demand of chicken has led to the rapid growth of poultry with the possible misuse of antibiotics. The paucity of information on the prevalence of resistance and MDR among enterococci in the environment has motivated the present study. In this context, this study was conducted in order to provide baseline information with the aim of determining the antibiogram profile and the rate of MDR among enterococci recovered from a poultry farm environment in Selangor state, Malaysia.

## 2. Results

A total of 80 enterococcal isolates were recovered from the environment of 28 poultry farms in Selangor with 56.3% belonging to *E. faecalis* (45/80) and 43.8% belonging to *E. faecium* (35/80). The sources of the enterococcal isolates were soil (28 farms) and effluent water (21 farms), with distribution of 57.5% (46/80) and 42.5% (34/80), respectively.

### 2.1. Antibiotic Resistance and Antibiograms of Enterococcal Isolates

Among the enterococcal isolates, 2.5% (2/80) were found to be susceptible towards all antibiotics tested. Whereas six isolates exhibited susceptibility towards nine antibiotics, with one intermediate resistance. Furthermore, 72 isolates (90%) were resistant to at least one antibiotic that was tested.

Table 1 shows the antibiogram of the enterococcal isolates recovered from the poultry environment in Selangor. Higher resistance rates (more than 50%) were observed among the enterococcal isolates towards tetracycline and erythromycin, followed by moderate levels of resistance (25 to 49%) towards linezolid and high-level streptomycin. All isolates were susceptible to tigecycline. Lower resistance rates (less than 25%) were reported towards ciprofloxacin, high-level gentamycin, ampicillin, vancomycin and teicoplanin.

The distribution of the resistance towards each tested antibiotic among the enterococcal isolates recovered from the poultry environment with respect to species is as shown in Figure 1. *E. faecium* exhibited significantly higher resistance compared to *E. faecalis* towards erythromycin, linezolid and high-level gentamicin.

Table 2 shows the antibiotic resistance profiling of the enterococcal isolates recovered from the poultry farm environment. Among 46 enterococcal isolates that exhibited multi-antibiotic resistance, 17 profiles were discovered. The most common antibiotic resistance profile in present study was TET/ERY/LZD and ten enterococcal isolates exhibited such profile. Erythromycin resistance was present in all phenotypes of multi-antibiotic resistance.

Linezolid-resistant enterococci (LRE) were observed only among multi-antibiotic-resistant isolates. All LRE co-existed with tetracycline and erythromycin resistance. Vancomycin exhibited co-resistance towards erythromycin and teicoplanin. Meanwhile, two isolates were revealed to be linezolid-resistant, vancomycin-resistant enterococci (LRVRE).

### 2.2. Multidrug Resistance (MDR) among the Enterococcal Isolates Recovered from the Poultry Farm Environment

MDR, which implies resistance towards three or more classes of antibiotics, was found to be present in 56.3% (45/80) of the enterococcal isolates. Among eight classes of antibiotics tested, a higher proportion of MDR was observed among the enterococcal isolates that were resistant towards four (21, 26%) and three (17, 21.3%) antibiotic classes. Meanwhile, approximately six isolates (7.5%) were resistant towards five classes of antibiotics. All linezolid- and ampicillin-resistant enterococci were found to be MDR. Higher MDR was found among *E. faecium* (69%) compared to *E. faecalis* (49%) isolates, with one isolate of *E. faecium* isolate possessing extensive drug resistance (XDR) and exhibiting resistance to seven of the tested classes of antibiotics. The term XDR is used to refer to isolates with resistance to at least one antibiotic among all except two or fewer categories to which it remains susceptible [12].

## 3. Discussion

Environments in which animal husbandry takes place are a reservoir for resistant bacteria and play an important role in their dissemination. In our study, *E. faecalis* was more predominant compared to *E. faecium*. A similar phenomenon was observed among clinical samples obtained in a hospital in Malaysia [13] and in raw poultry meat samples from Canada, Colombia and Sweden [14]. In contrast, studies conducted in Indonesia, Thailand, and Vietnam reported *E. faecium* to be the predominat species in samples of chicken feces [15]. The high resistance rate exhibited among enterococcal isolates from poultry farm environments are a source of concern, since the majority of the antibiotics tested in this study have been classified as ‘critically important antimicrobials for human medicine’ by the World Health Organization (WHO) [16]. Nevertheless, a notable and statistically significantly higher resistance towards erythromycin, linezolid and high level gentamycin was observed in *E. faecium* compared to *E. faecalis* in the present study.

Linezolid, as one of the last resorts for the treatment of enterococcal infection among humans, was included in the WHO list of critically important antibiotics [17,18]. In the midst of the emergence of multi-antibiotic resistance, linezolid, as a synthetic drug that belongs to the oxazolidinone class, was approved by the Food and Drug Administration (FDA) for clinical use in 2001 [19,20]. However, linezolid-resistant enterococci (LRE) have been reported since the commencement of its clinical use [8,19], and have been found to be quite predominant among *E. faecalis* in Asia [19]. These contradictory findings may be the result of differences in the domains in previous studies, which involved clinical isolates [19]. Despite a lower proportion of LRE among *E. faecalis* in the present study, the resistance rate was considered higher (28.9%) than that in a study involving fecal samples from broiler farms conducted from 2016 to 2018 in Korea [8].

This study found high resistance rates against tetracycline and erythromycin. Although we are aware that these antibiotics are no longer the first-choice treatment for enterococcal infection [14,21], the antibiotic resistance gene could spread interspecies between enterococci and staphylococci via the conjugation system through pheromone signaling [22]. Hence, tetracycline and erythromycin resistance should be taken into consideration. In many countries, tetracycline is added as a growth promoter through the livestock’s feed [23]. The usage of tylosin in animal husbandry as a growth promoter has also been linked to erythromycin-resistant enterococci [24]. A high concentration of tylosin was detected in the broiler manure obtained at almost all of the study sites in Malaysia [25]. However, there are no regulations on the use of those antibiotics in chicken feed during the sampling frame. Malaysia has adopted a system of phasing out by stages in order to achieve the prohibition of antibiotics as a growth promoter. Antibiotics being considered next to be banned include erythromycin, fosfomycin, tylosin, tilmicosin and neomycin [26].

A moderate level of resistance was observed among high-level streptomycin in the present study. Although there is bactericidal activity among enterococci towards aminoglycosides, those antibiotics can be used together with cell wall inhibitors, such as ampicillin, for a synergistic effect, achieving a better outcome in the treatment of enterococcal infection [27]. Moreover, such combinations were considered in the treatment of infective endocarditis caused by *E. faecalis* [28]. However, resistance to a high level of those aminoglycosides obliterates the synergistic effect [29]. We found a lower resistance rate to ciprofloxacins (20.0%) and high-level gentamicin (17.5%) in the present study. A study evaluating the VITEK 2 system among enterococcal species found 1.1% instances of very major error (*n* = 1/89) in which the isolate was reported to be susceptible to the system, in contrast to the reference method, which exhibited resistance [30]. Although we did not quantify antibiotic residues in our study, another local study did, and reported the presence of enrofloxacin, flumequine and norfloxacin residues, which may have contributed to the ciprofloxacin resistance found in all of the sampled manure obtained from broiler farms in Malaysia [25,31].

One of the most interesting observations emerging from our data was the low resistance rate to vancomycin (3.8%). A similar trend has been observed in other local studies [32,33]. A lower prevalence of VRE in the poultry environment could be corroborated on the basis of the banning of avorparcin and vancomycin in animal feed in this country, since both antibiotics, which are similar in terms of chemical structure, were linked to VRE following their use in feed for livestock [32,34,35]. Even, other studies have reported a reduction in vancomycin resistance following the prohibition of its usage in the veterinary sector [34]. Despite the low rate of vancomycin resistance, this finding is important and needs to be scrutinized. Such findings can be explained by the judicious use of vancomycin. In awareness of VRE and with concern for the limited number of therapeutic choices, its use is considered only as one of the last options for the treatment of enterococcal infection [34]. Moreover, teicoplanin, which is a glycopeptide [36], was also found to co-exist with VRE with very low resistance rates in the current study. Although we did not explore the molecular characterization of these isolates further, the mechanism mediating vancomycin and teicoplanin can be encoded by *vanA*, which is a cross-resistance gene [37].

We found that all LRE isolates in our study possessed MDR. A possible explanation for this could be information on the acquisition of transferable ribosomal protection genes like *optrA* and *poxtA* giving rise to resistance to linezolid, as well as conferring resistance to several other classes of anti-ribosomal antibiotics [38]. This was shown by another study that involved analysis of raw and ready-to-eat food samples of animal origin. It was reported that 96.4% of LRE were MDR, and found the *poxtA* gene to be the principal gene encoding LRE. The high rate of resistance LRE and MDR of should be viewed seriously, as it will pose a challenge to clinicians when there is a spread to humans [38]. Therefore, it is necessary for the relevant agencies to investigate the usage of linezolid in animal husbandry as well as any other antibiotics that can facilitate the development of LRE and to control the development of further resistance towards it.

A previous study evaluating the effectiveness of the VITEK 2 sytem for the identification and AST of Gram-positive cocci, reported that the system complies to the ‘minimal performance characteristics’ set by the Food and Drug Administration for antimicrobial susceptibility tests and provided accurate results. However, very major errors were reported for high-level gentamicin (1.1%, 1/89) and teicoplanin (4.5%, 4/89), with a categorical agreement of 99% and 91%, respectively, with a reference method for enterococcal species. The very major error implied in this context is that the VITEK 2 system reported the isolate to be susceptible to the tested antibiotic, in contrast to the reference method, which exhibited resistance [30]. Therefore, considering the errors reported previously, it is suggested to perform molecular testing for future studies. 

Among the limitations of our study is that we only studied the poultry environment alone in determining the rate of antibiotic resistance. However, this serves as a baseline and could instigate others to explore other fields of livestock husbandry. We also used simple random sampling, which provided an equal chance for each of the poultry farms listed under the Department of Veterinary Services (DVS) to be chosen from the registered. However, this approach is prone to selection bias, as farms that are not registered with DVS were not included in the study, and they could be considered in future studies.

## 4. Materials and Methods

### 4.1. Study Design and Source of Sampling

This cross-sectional study concentrated on antibiotic resistance of enterococcal species, which was a part of a larger project that was approved by the National Medical Research Register, Ministry of Health, Malaysia (NMRR-17-1198-36521). The whole project involved a variety of bacteria recovered from the environment of 33 poultry farms in Selangor state, Malaysia. The poultry farms were randomly chosen from the registry list provided by DVS. The study commenced in January 2018 and continued until October 2019. 

Samples taken included soil and/or effluent water from the respective farms. Among the 99 soil samples and 51 effluent water samples were collected, 80 enterococcal isolates were recovered from 28 out of 33 poultry farm environment. Analysis of antibiotic resistance was performed on *E. faecalis* and *E. faecium*. All of the laboratory work was conducted at the Institute of Medical Research (IMR).

### 4.2. Sample Collection, Preparation, and Culture

In each of the poultry farms, 25 g of soil samples were collected from three different areas that were 10 to 20 m away from one another. Soil samples were collected either from the coop area or at the location where the chicken flock clusters for open type farms. Meanwhile, random soil sampling was performed near the broiler chicken house for the closed type farms. The top layer of soil to a depth of 3 cm was removed, using a metal spade to collect the sample.

Meanwhile, a long-handled stainless steel ladle was used to collect 200 mL of effluent water based on the availability of a stagnant pool of water within the farm or drainage system. The spade and ladle were cleaned with water, disinfected with 75% alcohol and heated with flame using a Bunsen burner prior to the collection of each sample. The collected samples were kept in sterile zip-lock plastic bags and transported via cool box to a microbiology laboratory for further analysis.

The soil and effluent samples from each farm were pooled in a sterile zip-lock plastic bag. The soil samples were homogenized using a spatula. Meanwhile, mixing was performed by shaking the bag up and down and side to side for the effluent water. Ten grams of soil was weighed, and added into 90 mL of peptone water. The mixture was then vortexed. 

Subsequently, 1 mL of the aliquot was transferred into a tube containing 9 mL of peptone water and vortexed. The process was repeated for subsequent dilution until 10-fold water to reduce the bacterial concentration for further analysis. A similar method was used for 10 mL of effluent water.

Following this, 1 mL from each dilution tube was poured onto a commercially prepared HiMedia *Enterococcus* Differential (ECODI) agar plate, which was stored at 4 °C until use. A sterile, disposable spreader was used to disperse the aliquot uniformly on the agar. Then, the agar plates were incubated for 24 h at 34 °C. The color of the colonies of *E. faecalis* and *E. faecium* were red or maroon and colorless, respectively [39].

A minimum of three representative colonies per plate were selected and subcultured twice successively onto Trypticase Soy Agar (TSA) for identification of the bacteria and antibiotic susceptibility testing. However, if isolates of the same species from the same origin exhibited indistinguishable resistance patterns, only one isolate was randomly included in this study. 

### 4.3. Identification and Antibiotic Susceptibility Testing

The VITEK 2 system (bioMérieux), an automated machine that works of the basis of fluorescence technology is shown in Figure 2, and was used to identify the bacteria and determine susceptibility towards antibiotics. The principle of the system was elaborated in detail by Funke and Funke-Kissling [40]. Version 8.01 software was used.

Gram staining was performed for all recovered isolates to determine Gram reaction and morphology of the bacteria. The isolates that were Gram positive proceeded further for identification of the bacteria using VITEK^®^2 Gram-Positive Identification cards (GP-ID) (bioMérieux, Nurtingen, Germany). After identification of enterococcal species, we proceeded with antibiotic susceptibility testing using AST-P592 cards (bioMérieux), which are specifically for *Staphylococcus* spp., *Enterococcus* spp. and *Streptococcus agalactiae* [41].

Among the 22 antibiotics, 10 were reported for *enterococcus spp.*, belonging to eight group of antibiotics: (1) penicillin: ampicillin (AMP); (2) aminoglycosides: high-level gentamicin (HI-GEN), streptomycin (STR); (3) fluroquinolones: ciprofloxacin (CIP); (4) macrolides: erythromycin (ERY); (5) oxazolidinones: linezolid (LZD); (6) glycopeptides: teicoplanin (TEC), vancomycin (VAN); (7) tetracycline (TET); (8) glycylcyclines: tigecycline (TGC).

The results of antimicrobial susceptibility testing (AST) as generated by the VITEK system were based on bacterial growth in the wells of the AST card in comparison to a positive control well, and were reported automatically by the system in accordance with the stipulations of the Global Clinical and Laboratory Standards Institute (CLSI) in 2017. The reported results, which were based on an algorithm generated by VITEK, were equivalent to actual minimal inhibitory concentration (MIC). No further test was performed on the standard phenotype for actual MIC determination (such as broth microdilution or agar dilution techniques) and resistance genes in the present study. As a control strain, *Enterococcus Faecalis* ATCC 29212 was used. The isolates were further identified manually for MDR and XDR.

### 4.4. Statistical Analysis

Statistical Package for Social Sciences (SPSS) IBM software version 27 was used for statistical analyses. Chi-squared test was used to determine whether the distribution of resistance to antibiotics differed between the two enterococcal species. *p*-values less than 0.05 were considered to be significant results. 

## 5. Conclusions

In conclusion, enterococcal isolates recovered from a poultry farm environment exhibited multi-resistance towards critically important antibiotics for human use. This is a worrying situation, because the environment is becoming a reservoir harboring resistant bacteria and genes, which could be disseminated to humans and animals. In line with the “One Health Concept”, findings in the present study provide evidence of the growing antimicrobial resistance and its pattern in poultry farm environments. These findings should be taken seriously by relevant agencies in an effort to strengthen collaborative public health interventions with the aim of controlling and preventing further resistance. This study also identified future research that could be carried out, such as quantifying the concentration of antibiotic residues in the environment along with performing the molecular characterization of the isolates. This will provide insights into the causes of resistance and its patterns, and will further aid in developing the intended interventions.

## Figures and Tables

**Figure 1 antibiotics-11-01118-f001:**
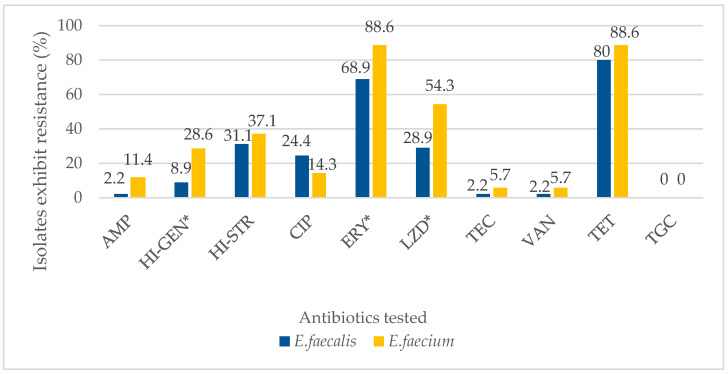
Rates of antibiotic resistance among *E. faecalis* and *E. faecium* isolates recovered from a poultry farm environment in Selangor. (AMP: ampicillin; HI-GEN: high-level gentamicin; HI-STR: ligh-level streptomycin; CIP: ciprofloxacin; ERY: erythromycin; LZD: linezolid; TEC: teicoplanin; VAN: vancomycin; TET: tetracycline; TGC: tigecycline; 1: *E. faecalis*; 2: *E. faecium*). * There were significant differences between *E. faecalis* and *E. faecium* (*p* < 0.05).

**Figure 2 antibiotics-11-01118-f002:**
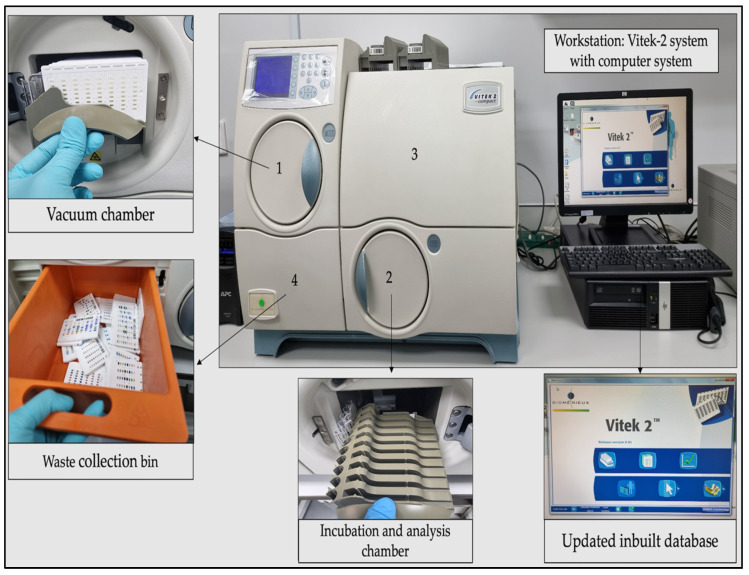
Workstation VITEK 2 system with computer system. Note: 1—fill door; 2—load door; 3—front user access door; 4—waste collection door.

**Table 1 antibiotics-11-01118-t001:** Antibiogram of enterococcal isolates recovered from a poultry farm environment in Selangor.

Antibiotics Tested	Number of Isolates Tested	Antibiogram of Enterococcal Isolates		
		Resistant (%)	Intermediate (%)	Susceptible (%)
Ampicillin	80	5 (6.3)	0	75 (93.8)
High-level gentamicin	80	14 (17.5)	0	66 (82.5)
High-level Streptomycin	80	27 (33.8)	1 (1.3)	52 (65.0)
Ciprofloxacin	80	16 (20.0)	22 (27.5)	42 (52.5)
Erythromycin	80	62 (77.5)	14 (17.5)	4 (5.0)
Linezolid	80	32 (40.0)	12 (15.0)	36 (45.0)
Teicoplanin	80	3 (3.8)	0	77 (96.3)
Vancomycin	80	3 (3.8)	1 (1.3)	76 (95.0)
Tetracycline	80	67 (83.8)	0	13 (16.3)
Tigecycline	80	0	0	80 (100)

**Table 2 antibiotics-11-01118-t002:** Antibiotic resistance profiling of *E. faecalis* and *E. faecium* recovered from a poultry farm environment.

Number of Antibiotics	Phenotypes	Antibiotic Resistance Profile	*E. faecalis* (*n* = 45)	*E. faecium* (*n* = 35)
Resistance to 3 or more antibiotics				
7	P1	TET/ERY/LZD/CIP/AMP/VAN/TEC	NA	1
6	P2	TET/ERY/LZD/AMP/VAN/TEC	1	NA
	P3	TET/ERY/LZD/HI-STR/CIP/HI-GEN	NA	2
5	P4	TET/ERY/LZD/HI-STR/CIP	1	1
	P5	TET/ERY/LZD/HI-STR/HI-GEN	NA	4
	P6	TET/ERY/LZD/CIP/HI-GEN	1	NA
	P7	TET/ERY/HI-STR/CIP/HI-GEN	2	NA
	P8	TET/ERY/HI-STR/HI-GEN/AMP	NA	1
4	P9	TET/ERY/LZD/HI-STR	4	3
	P10	TET/ERY/LZD/HI-GEN	NA	2
	P11	TET/ERY/LZD/AMP	NA	2
	P12	TET/ERY/HI-STR/CIP	2	NA
	P13	TET/ERY/CIP/HI-GEN	1	NA
3	P14	TET/ERY/LZD	6	4
	P15	TET/ERY/HI-STR	2	2
	P16	TET/ERY/CIP	2	1
	P17	ERY/VAN/TEC	NA	1
Resistance to one or two antibiotics				
2	P18	TET/ERY	7	6
	P19	TET/HI-STR	2	NA
	P20	ERY/HI-GEN	NA	1
	P21	ERY/CIP	2	NA
	P19	TET/HI-STR	2	NA
1	P22	TET	5	2
	P23	HI-STR	1	NA

NA: Not available.

## Data Availability

The data is contained within the article.

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
