# Peer review of "Resistance towards Critically Important Antimicrobials among Enterococcus faecalis and E. faecium in Poultry Farm Environments in Selangor, Malaysia"

_antibiotics, 2022, doi:10.3390/antibiotics11081118_

Round 1
Reviewer 1 Report
A minor revision is required:
1) Line 63-64: E. faecalis (56.3%, 45/80) and E. faecium (43.8%, 35/80) were identified from soil, (57.5%, 46/80) and effluent water (42.5%,34/80).
Question: - Are these parts from "(57.5%, 46/80) and (42.5%, 34/80)" were E. faecalis and E. faecium identified from effluent water? If so, change to : (57.5%, 46/80) and (42.5%, 34/80) from effluent water, respectively.
2). line 67-69 of the results part: It describes that among the enteroccal isolates, 2.5% (2/80) were found to be susceptible towards all antibiotics tested. It was followed by isolates exhibiting susceptibility toward nine antibiotics with one intermediate resistance.
Comment: - These two sentences conflicted with the data shown in the table1. There are no described data shown in the table 1.
3). Line 84-85: Figure1. It is a confused word between E. faecium and E.faecium itself.
Comment: Based on the data depicted in the figure 1 shown the E. faecium exhibited significantly higher resistance than E. faecalis towards ERY, LZD and HI-GEN.
4). In statistical analysis part:
- Comment: It will be comprehensive if it can elaborated a little bit about the Chi-square test on what variables to test for. In addition, there is also no any statement on the statistical significance association of variables in the discussion part although it has been set for the level of significance of P-value less than 0.05.
Reviewer 2 Report
The manuscript entitled “Prevalence of Enterococcal Resistance Towards Critically Important Antimicrobials from Poultry Farm Environment in Selangor, Malaysia” present isolation of Enterococcus faecalis and E. faecium from soil and effluent water in poultry farm environment and antimicrobial resistance with a focusing on linezolid-resistance enterococci associated with multidrug resistance phenotypes. The rationale and introduction are clear. However, there are some gaps of methodologies that need to be improved, and some points need to be re-written in the materials and methods, results and discussion. Please see the comments below.
Line 2-3. The manuscript title should be revised, for example, “Linezolid-resistant Enterococcus faecalis and E. faecium from Poultry Farm Environment in Selangor, Malaysia.” The terminology “prevalence” should be avoided if the samples was not calculated or well designed in term of epidemiological surveillance. However, rigid methodology by standard methods of identification of linezolid resistance need to be added (see below).
Line 62-64. The authors should provide the data about how many farm were positive? What kind(s) of sample is positive in each farm?
Line 66. Please use “antibiograms” or “antimicrobial resistance profiles”
Line 72. “High rates” or “high frequencies” of resistance are suggested instead of “High prevalence”.
Line 73-74. “Tigecycline was 100% effective against the enterococcal isolates” should be re-written because this is not evaluation of treatment of infection. For instance, “All isolates are susceptible to tigecycline” is suggested.
Line 87. “Rates of resistance” should be used instead of “prevalence”.
Line 158. Intrinsic resistance to aminoglycosides in enterococci is low permeability. “due to lack of bactericidal activity” is the result from intrinsic resistance, not the cause.
Line 168-169. The sentence is not clear about the low rate of vancomycin resistance. The interesting is either low rate or the presence of vancomycin resistance. The presence of vancomycin, even in low frequency, should be considered as importance.
Line 180, 182, 186. The gene nomenclature needs to be corrected. Please use italic for the gene names and lowercase for the first character of the name, for example, vanA (italic), optrA (italic), and poxtA (italic).
Line 180. The mechanism mediating vancomycin and teicoplanin (glycopeptides) can be encoded by vanA that is a “cross-resistance”.
Line 243-267. These paragraphs should be shortened because this is the protocol from manufacturer that is not necessary to include in the manuscript.
Line 268-271. The MIC results from Vitek automated system are not the actual MIC because the system is not based on standard broth dilution or microdilution assay. The results from Vitek antimicrobial susceptibility testing (AST) from bacterial growth in the wells of AST card are only equivalent to actual MIC by the algorithm generated by Vitek system.
In this regard, in case of resistance critically important antimicrobials such as linezolid and vancomycin, standard phenotypic test by actual MIC determination (by broth microdilution or agar dilution techniques) and/or resistance gene detection should be further performed.
